# Clinical characteristics and outcomes of intraocular lens dislocation: an eight-year retrospective study

**Su Xu, Jingzhi Shao, Yuhang Zhang, Wei Si, Yi Mao, Shanshan Du, Fengyan Zhang** *

Department of Ophthalmology, The First Affiliated Hospital of Zhengzhou University, Zhengzhou University, Zhengzhou, Henan, China

* zhangfengyanx@aliyun.com

## Abstract

### Objective

To investigate the clinical characteristics, risk factors, and short-term visual outcomes following surgical management of intraocular lens (IOL) dislocation, aiming to inform risk stratification and individualized treatment strategies.

### Methods

A retrospective analysis was conducted on 155 patients (166 eyes) diagnosed with IOL dislocation between 2016 and 2024. Data collected included demographic information, ocular and systemic risk factors, surgical interventions, uncorrected distance visual acuity (UDVA) before and after surgery, and postoperative complications. Comparative analyses were performed to assess the short-term efficacy and safety of different surgical approaches.

### Results

The mean patient age was 55 years, with 86.75% presenting unilateral dislocation. Predominant risk factors identified were high myopia (38.55%) and prior vitrectomy (17.47%). A total of 89.16% of IOL dislocations were classified as late-onset. Of the 154 eyes that underwent surgical intervention, both IOL repositioning and exchange procedures resulted in significant improvements in UDVA ($P < 0.05$). Postoperative transient intraocular pressure elevations were observed without significant differences between surgical methods, resolving within three days in all cases.

### Conclusion

High myopia and prior vitrectomy are significant risk factors for IOL dislocation. Surgical correction, whether through repositioning or exchange, is effective and safe, leading to substantial visual acuity improvements. Proactive identification and

**Data availability statement:** All relevant data are within the paper and its Supporting Information files.

**Funding:** The author(s) received no specific funding for this work.

**Competing interests:** The authors have declared that no competing interests exist.

management of high-risk individuals are crucial for preventing IOL dislocation and optimizing patient outcomes.

---

## 1 Introduction

With the increasing adoption of cataract phacoemulsification and intraocular lens (IOL) implantation, the incidence of IOL dislocation has become more common, with reported rates ranging from 0.1% to 3%. IOL dislocation can lead to visual impairment and other ocular symptoms, often requiring additional surgery to restore vision, which increases both the economic burden and medical risk for patients. As a result, research on the prevention, early diagnosis, and individualized surgical management of IOL dislocation remains a key focus in ophthalmology.

Studies have shown that high myopia, uveitis, pseudoexfoliation syndrome, retinitis pigmentosa, and a history of intraocular surgery may be risk factors for late-onset IOL dislocation. Intraoperative complications during primary cataract surgery, such as a small continuous curvilinear capsulorhexis, capsular rupture, or zonular damage, are also considered important causes of early dislocation.

Despite these findings, studies addressing regional differences in clinical characteristics and risk factors are limited. There is also a lack of analysis regarding visual recovery and the effectiveness of different surgical approaches. This study retrospectively analyzes clinical data from patients with IOL dislocation in the Central Plains region of China, focusing on clinical features, risk factors, and postoperative visual outcomes. The goal is to provide evidence to support early identification and personalized treatment strategies for high-risk patients.

## 2 Methods

### 2.1 Study design and participants

This was a retrospective observational study. A total of 155 patients (166 eyes) diagnosed with IOL dislocation between January 2016 and October 2024 were included.
Inclusion criteria:

(1) History of IOL implantation;

(2) Dislocation of the IOL or the IOL–capsular bag complex, confirmed by slit-lamp biomicroscopy, ophthalmoscopy, ultrasound biomicroscopy (UBM), or optical coherence tomography (OCT), defined as dislocation >0.5 mm from the original position or tilt angle >7° following adequate mydriasis with tropicamide;

(3) Complete clinical and follow-up data, including detailed preoperative, intraoperative, and postoperative records, and at least one day of postoperative follow-up;

(4) Normal cognitive function and the ability to cooperate with ocular examinations.

Exclusion criteria:

(1) Severe corneal pathology, endophthalmitis, or other uncontrolled ocular conditions preventing necessary examinations;

(2)  Incomplete clinical data, inadequate postoperative follow-up, or failure to comply with follow-up plans resulting in missing key pre- or postoperative data.

All patient records and data were anonymized and de-identified. The study was approved by the institutional ethics committee (2022-KY-0006–001), and written informed consent was obtained from all participants. In the case of minors, written consent was obtained from their guardians.

## 2.2  Observational parameters

Data collected included demographics, characteristics of the dislocation, systemic and ocular comorbidities, surgical methods, uncorrected distance visual acuity (UDVA), postoperative intraocular pressure (IOP), and complications.

Hypertension was defined according to JNC8 guidelines as blood pressure ≥140/90 mmHg or current use of antihypertensive medication [1]. Diabetes mellitus was defined by ADA criteria as fasting plasma glucose ≥126 mg/dL, random plasma glucose ≥200 mg/dL, or HbA1c ≥ 6.5% [2].

High myopia was defined as axial length ≥25 mm. Other comorbidities were identified based on medical records. Particular attention was given to details of the initial cataract surgery, including continuous curvilinear capsulorhexis (CCC) diameter <5.0 mm, capsular rupture, zonular dialysis or laxity, and use of iris hooks or capsular tension rings.

## 2.3  Ocular examinations

After full pupil dilation with tropicamide, slit-lamp biomicroscopy was used to observe and photograph IOL dislocation and associated signs such as nystagmus, iridodonesis, and corneal edema. AL and IOL power were measured using the IOL Master. Non-contact tonometry was used to measure IOP. UBM was employed to assess the IOL position, zonular integrity, ciliary body, and anterior chamber angle. B-scan ultrasonography, fundus photography, and OCT were used to examine the vitreous, retina, macula, and optic disc.

## 2.4  Surgical techniques

**2.4.1  IOL repositioning.** ① In-the-bag repositioning: If the IOL remained adjustable intraoperatively, a 3.0 mm clear corneal incision was made at the 10:30 position under satisfactory anesthesia. Viscoelastic material was injected into the anterior chamber, and a 1.0 mm side port was created at the 2:00 limbus. A dialing hook was used to reposition the dislocated IOL into the capsular bag. After repositioning, the viscoelastic was removed, and the incision was closed by hydration or suturing.

② Intrascleral fixation: For unstable capsular support or high risk of re-dislocation, scleral fixation was performed. The sclera at the 2:00 and 8:00 positions was exposed. An 8−0 polypropylene suture was passed through the sclera 2 mm posterior to the limbus at 8:00, entering the pupillary zone. A 29-gauge needle was introduced at the 2:00 limbus to dock the suture, which was externalized. The suture ends were pulled through the main corneal incision, trimmed, and secured to the IOL haptics. The IOL was then fixated in a Z-suture technique to the scleral layers. The corneal incision was closed with 10−0 nylon sutures, and the conjunctiva was closed with 8−0 absorbable sutures [3] (Fig 1).

**2.4.2  IOL exchange.**  Indicated for patients with unsuitable IOLs for scleral fixation or with refractive power errors. The original IOL was extracted through a limbal incision at the 10:30 position. Depending on the condition of the capsulorhexis, the new IOL was implanted using either anterior capsular optic capture or scleral fixation techniques, as described above. For reimplanted IOLs, all preset powers were 0°.

**2.4.3  IOL explantation.**  Performed in cases where ocular conditions precluded re-implantation of an IOL.

In all surgical methods, anterior vitrectomy was conducted if vitreous prolapse into the anterior chamber was observed intraoperatively. Pars plana vitrectomy was combined when the IOL had dislocated into the vitreous cavity.

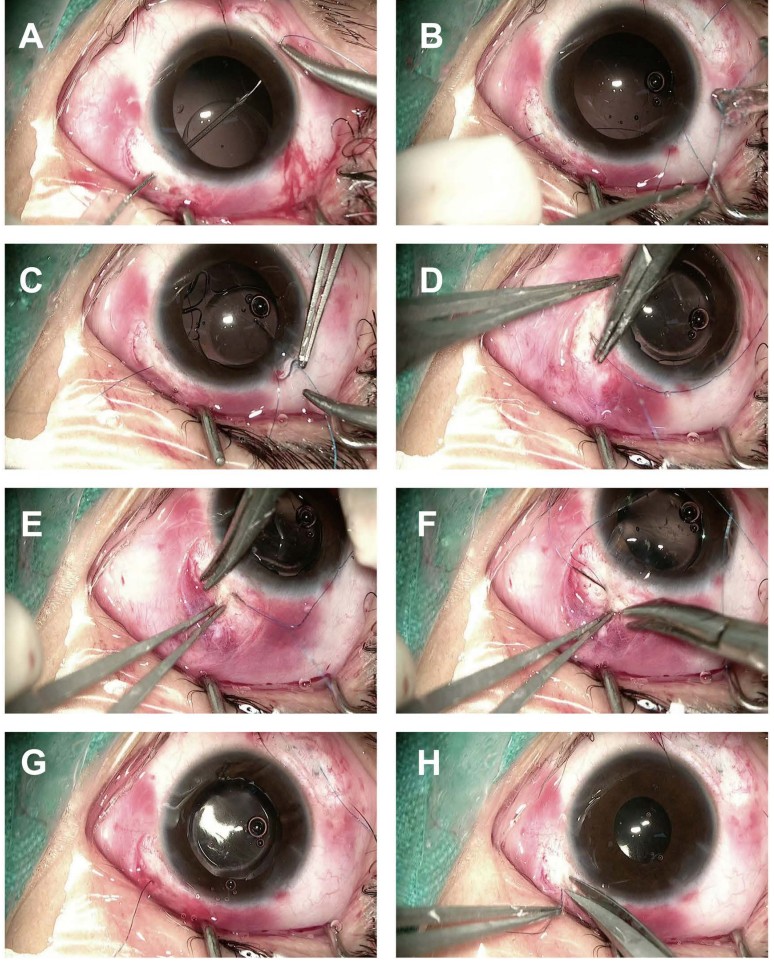

**Fig 1. Schematic illustration of scleral fixation of the IOL. A.** Docking of the 8−0 polypropylene suture needle with a 29-gauge needle; **B.** The externalized suture end at the 2:00 position is passed through one closed haptic of the IOL and tied; **C.** The suture end at the 8:00 position is passed through the diagonally opposite closed haptic of the IOL; **D–H.** Z-shaped intrascleral fixation technique [3].

## 2.5 Statistical analysis

Statistical analysis was performed using SPSS version 27.0. Data distribution was assessed using the Kolmogorov–Smirnov or Shapiro–Wilk test. Normally distributed continuous variables were expressed as mean ± standard deviation (SD) and compared using independent samples $t$-tests. Non-normally distributed data were expressed as median and interquartile range (IQR), and compared using nonparametric rank-sum tests. Categorical variables were presented as frequencies and percentages and compared using the $\chi^2$ test. A $P$-value <0.05 was considered statistically significant.

## 3 Result

### 3.1 Sociodemographic characteristics

A total of 155 patients (166 eyes) were included in this study, comprising 114 males (122 eyes) and 41 females (44 eyes), with a median age of 55 years (interquartile range: 37–64; range: 6–87 years). There was no significant difference in the age of onset between males and females ($Z = -0.260$, $P = 0.795$). The peak onset age for males was between 49

and 59 years, while for females it was between 59 and 69 years, indicating an earlier peak in males. Among all patients, 65 (41.94%) resided in rural areas and 90 (58.06%) lived in urban areas. No significant difference was observed in the urban-rural distribution.

### 3.2 Comorbidities

Of the 155 patients, 44 had hypertension and 16 had diabetes. High myopia was present in 64 cases (38.55%), followed by a history of pars plana vitrectomy (PPV) in 29 cases (17.47%) (Table 1).

### 3.3 Personal history of unhealthy habits

A total of 41 patients (26.45%) reported unhealthy habits, including 33 (21.29%) with a smoking history ranging from 1 to 50 years (mean: 26.73±13.03 years) and an average daily consumption of 16.85±8.81 cigarettes. Eight patients (5.16%) were heavy drinkers with a drinking history of 10–50 years (mean: 28.75±13.56 years) and an average daily alcohol intake of 112.50±44.32 mL.

### 3.4 Characteristics of IOL dislocation

Among the 155 patients, 166 eyes exhibited IOL dislocation, with unilateral dislocation in 144 patients (144 eyes, 86.75%) and bilateral dislocation in 11 patients (22 eyes, 13.25%). Fifteen eyes (9.04%) experienced recurrent IOL dislocation. The time from the initial surgery to dislocation varied widely—from as early as 1 day postoperatively to up to 20 years. Dislocation occurred within 3 months in 18 cases (10.84%) and beyond 3 months in 148 cases (89.16%), with a mean interval of 7.57±6.48 years (range: 1 day to 25 years).

Prior to dislocation, 24 patients (14.46%) reported a clear triggering factor, including trauma (15 cases, 9.04%), eye rubbing (7 cases, 4.22%), and heavy lifting (2 cases, 1.20%). After dislocation, 4 patients (2.41%) were asymptomatic, and dislocation was detected incidentally during follow-up. Ocular symptoms were reported in 162 eyes (97.59%), most commonly blurred vision alone (129 eyes, 77.71%). Other symptoms included visual instability, eye pain, photophobia,

**Table 1. Systemic and ocular comorbidity in patients with IOL dislocation*.**

| Disease | n(%) |
|---|---|
| **Systemic diseases** | |
| Hypertension | 44(28.38) |
| Diabetes | 16(10.32) |
| Atopic diseases (allergy, asthma) | 13(8.39) |
| Post-cholecystectomy | 7(4.52) |
| Post-coronary stent implantation | 5(3.23) |
| **Ocular diseases** | |
| High myopia | 64(38.55) |
| Prior vitrectomy | 29(17.47) |
| Previous acute angle-closure glaucoma | 8(4.82) |
| Macular epiretinal membrane | 6(3.61) |
| Capsular contraction syndrome | 6(3.61) |
| Posterior capsular opacification | 6(3.61) |
| Amblyopia | 6(3.61) |
| Nystagmus | 5(3.01) |

* **Note:** Only conditions with a frequency ≥5 are included.

diplopia, and metamorphopsia (33 eyes, 19.88%). On examination, capsular fibrosis was observed in 44 cases (26.51%) and iris incarceration in 22 cases (13.25%). Detailed dislocation characteristics are shown in Fig 2 and Table 2.

### 3.5 Surgical treatments and outcomes

Of the 166 eyes, 154 underwent surgical treatment. The remaining 12 eyes did not receive surgery due to low endothelial cell counts (<500 cells/mm²) or patient concerns about surgical outcomes (Table 3). All surgeries were completed successfully without intraoperative complications.

UDVA significantly improved after both IOL repositioning and IOL exchange procedures ($P=0.001$ and $P=0.031$, respectively). Comparisons of preoperative and postoperative visual acuity across different surgical methods are shown in Table 4.

On postoperative day one, slit-lamp examination confirmed that the IOLs were well-centered, without decentration or tilt. The mean IOP was 15.50 mmHg (interquartile range: 11.75–19.00 mmHg; range: 7.00–49.00 mmHg). Elevated IOP was observed in 18 eyes (10 in the repositioning group and 8 in the exchange group), with no statistically significant difference in complication rates between the two groups ($\chi^2=2.751$, $P=0.097$). Elevated IOP was managed with anterior chamber paracentesis or intraocular pressure-lowering medications, and IOP returned to normal within 3 days in all cases.

## 4 Discussion

With the widespread adoption and increasing prevalence of phacoemulsification cataract surgery and IOL implantation, the incidence of partial and complete IOL dislocation has been rising in recent years. The reported incidence of IOL dislocation ranges from approximately 0.1% to 3.0% [5–7]. Once dislocation occurs, patients often experience significant visual impairment and face the risk of serious complications such as elevated intraocular pressure, retinal detachment, and vitreous hemorrhage [8–10,11]. Therefore, a thorough understanding of the clinical characteristics and risk factors associated with IOL dislocation is critical for early diagnosis and individualized intervention. Based on eight years of clinical data,

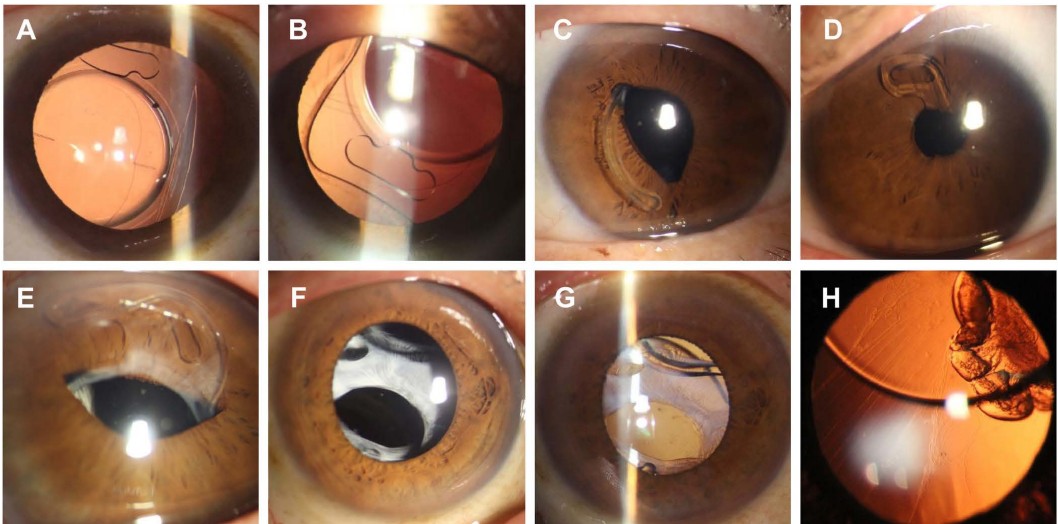

**Fig 2. Schematic illustrations of IOL dislocation. (A)** Dislocation of the IOL-bag complex due to capsular bag necrosis syndrome. **(B)** Supernasal dislocation of the IOL, with invisible anterior capsulorhexis margin. **(C-D)** The IOL is embedded in the pupil, causing pupil deformation. **(E)** The IOL-bag complex dislocates anteriorly, making contact with the corneal endothelium. **(F-G)** Capsular contraction causes downward dislocation of the IOL-bag complex, with the white organizing membrane of the anterior capsule visible. **(H)** Superior dislocation of the IOL, with the inferior zonule elongated and partially ruptured.

**Table 2. Characteristics of IOL dislocation.**

| Parameter | | | Eye(%) |
|---|---|---|---|
| **Number of eyes involved** | | | |
| Unilateral | Right | | 70(42.17) |
| | Left | | 74(44.58) |
| Bilateral | | | 22(13.25) |
| **Interval between implantation and dislocation** | | | |
| Within 3 months | | | 18(10.84) |
| After 3 months | | | 148(89.16) |
| **Inducement** | | | |
| With obvious inducement | Trauma | | 15(9.04) |
| | Eye rubbing | | 7(4.22) |
| | Heavy lifting | | 2(1.20) |
| Without obvious inducement | | | 142(85.54) |
| **Season of occurrence** | | | |
| Spring | | | 41(24.70) |
| Summer | | | 48(28.92) |
| Autumn | | | 47(28.31) |
| Winter | | | 30(18.07) |
| **Symptoms** | | | |
| With obvious symptoms | Blurred vision | | 129(77.71) |
| | Others | Diplopia | 10(6.02) |
| | | Metamorphopsia | 2(1.20) |
| | | Eye pain | 8(4.82) |
| | | Eye redness | 6(3.61) |
| | | Visual obstruction | 4(2.41) |
| | | Foreign body sensation | 2(1.20) |
| | | Dryness | 1(0.60) |
| Without obvious symptoms | | | 4(2.41) |
| **Complications** | | | |
| Organization of the capsule | | | 44(26.51) |
| Iris incarceration | | | 22(13.25) |
| Cystoid macular edema | | | 9(5.42) |
| Corneal edema | | | 2(1.20) |
| Inflammatory response | | | 1(0.60) |
| **Causes** | | | |
| Progressive zonule insufficiency | | | 138(83.13) |
| Suture degradation/breakage | | | 15(9.04) |
| Capsular contraction syndrome | | | 6(3.61) |
| Ruptured zonule | | | 3(1.81) |
| Ruptured capsule | | | 2(1.20) |
| IOL loop degradation/disruption | | | 2(1.20) |

this study provides a comprehensive summary of the clinical manifestations, risk factors, and short-term postoperative outcomes of patients with IOL dislocation, offering valuable guidance for clinical practice.

The average age of patients in this study was 55 years, approximately 15 years younger than reported in previous studies [12,13]. The high incidence of high myopia, prior vitrectomy, and younger age reflect the complexity and severity of IOL

**Table 3. Details of surgical treatments for IOL dislocation.**

| Parameter | eye(%) |
|---|---|
| **Surgical technique** | |
| IOL repositioning | 100(64.94) |
| IOL replacement | 39(25.32) |
| IOL explantation | 12(7.79) |
| Anterior capsular fibrotic membrane removal | 3(1.95) |
| **IOL fixation method** | |
| Single-loop fixation | 34(22.08) |
| Double-loop fixation | 115(74.68) |
| IOL-bag complex fixation | 5(3.25) |
| **Vitrectomy performed** | |
| Anterior vitrectomy | 30(19.48) |
| Complete vitrectomy | 23(14.94) |

**Table 4. Comparison of UDVA between different surgical techniques for IOL dislocation*.**

| Surgical technique | Preoperative UDVA | Postoperative UDVA | Z | P |
|---|---|---|---|---|
| IOL repositioning ($n = 100$) | 0.70 (0.30, 1.28) | 1.50 (1.00, 2.00) | −3.187 | 0.001 |
| IOL replacement ($n = 39$) | 0.40 (0.80, 1.30) | 1.50 (1.00, 2.00) | −2.159 | 0.031 |
| IOL explantation ($n = 12$) | 1.35 (0.94, 2.00) | 1.50 (1.00, 2.00) | −0.352 | 0.725 |

**\*Note:** UDVA is expressed as log MAR. Finger count: LogMAR 2.0; Hand move: LogMAR 3.0; Light Perception: LogMAR 4.0 [4].

dislocation cases. Gender analysis revealed a higher proportion of male patients, consistent with previous studies [14–16]. This gender difference may be attributable to both behavioral and biological factors. Trauma, a recognized cause of IOL dislocation, is more prevalent in males [17]. Another hypothesis is that males may exhibit more severe zonular weakness, though biological evidence remains limited, and further studies are required to confirm this [18]. This study further analyzed the differences in age at onset between male and female patients. Although no statistically significant difference was found, the peak age of onset occurred slightly earlier in males, suggesting the potential involvement of sex-related differences in the underlying pathological processes.

Risk factors associated with IOL dislocation include trauma (e.g., injury, repeated eye rubbing) [19–21], high myopia [9,22], prior vitrectomy [16,23,24], and previous episodes of acute glaucoma [25,26]. Among systemic conditions, hypertension, diabetes, atopic diseases, prior cholecystectomy, and coronary stent implantation were frequently observed. This suggests that these systemic diseases may affect intraocular structural stability through mechanisms such as chronic inflammation and hemodynamic alterations. The associations of diabetes [27] and atopic diseases [28] with IOL dislocation have been previously reported. Smoking, reported by one-fourth of patients in this study, is associated with structural changes in ocular tissues, such as retinal nerve fiber layer thinning [29–32] and elastic fiber damage in other organs [33], though its relationship with zonular integrity remains unexplored. This study suggests that smoking may be an under-recognized risk factor, warranting further investigation.

High myopia, noted in 64 cases (38.55%), was the most common ocular risk factor, consistent with prior studies [12,34,35]. Highly myopic eyes exhibit fundus changes (e.g., lacquer cracks, chorioretinal atrophy, posterior scleral staphyloma) and increased axial length, which exert additional stress on the zonules [36,37], increasing the risk of IOL dislocation. Prior vitrectomy was another significant risk factor [22], likely due to intraoperative zonular damage. Additionally, glaucoma was more prevalent in patients with IOL dislocation [38], potentially linked to sustained elevated IOP

causing zonular fiber degeneration, ciliary body dysfunction, and reduced capsular bag support [39]. These findings align with our results, highlighting the importance of zonular and capsular stability in maintaining the IOL position.

Based on the timing of onset, IOL dislocation can be classified as early (within three months after surgery) or late (three months or more postoperatively) [40]. In this study, the mean interval between IOL implantation and dislocation surgery was 7.57 years, which aligns with previously reported data (6.9–8.5 years) [40]. Early IOL dislocation is typically associated with capsular rupture or zonular damage caused by surgical trauma, whereas late dislocation is primarily attributed to progressive zonular insufficiency [40]. In this study, 89.16% of cases were classified as late dislocations, with progressive zonular insufficiency identified as the predominant cause. The zonular fibers, which are inserted into the equatorial lens capsule, bear a limited load [41]. Age-related weakening of these fibers exacerbates their fragility, and any disruption can lead to capsular instability, causing IOL displacement or prolapse [23]. This progressive age-related zonular dysfunction accounts for the time-dependent increase in late IOL dislocations [7,42]. Capsular contraction emerged as another significant factor contributing to IOL dislocation in this study. Contraction exerts additional stress on compromised zonular fibers, leading to dislocation. While mild capsular contraction commonly occurs after IOL implantation [43], it generally does not cause dislocation unless exacerbated by underlying conditions such as uveitis [44] or retinitis pigmentosa [27]. These conditions can lead to severe capsular contraction, known as capsular contraction syndrome(CCS), which places excessive strain on the already compromised zonule [19]. CCS is strongly associated with continuous curvilinear capsulorhexis. If the capsulorhexis diameter is too small, capsular fibrosis may occur [8,19], creating a "sphincter effect" around the anterior capsule opening, which intensifies capsular contraction [8]. This phenomenon is particularly severe in advanced IOL dislocation, where progressive zonular damage increases susceptibility to centripetal forces, raising the likelihood of dislocation [8]. The use of stronger sutures can mitigate suture breakage in certain cases. Replacing 10−0 polypropylene sutures with 9−0 polypropylene or 8−0 nonabsorbable sutures have been shown to reduce the incidence of suture degradation [45–47].

In this study, with the exception of 12 eyes that underwent conservative management due to specific conditions, all other cases received surgical intervention. Postoperatively, both IOL repositioning and IOL exchange procedures resulted in significant improvement in UDVA. The most common postoperative complication was transient IOP elevation, which was transient and manageable, consistent with previous studies [12,19]. This may be related to residual viscoelastic substances [48,49]. Notably, no cases of serious complications such as retinal detachment were observed, indicating a relatively high level of surgical safety.

Reports on the comparative outcomes of different surgical techniques have been inconsistent. Some previous studies suggest that IOL exchange may be more favorable than repositioning in terms of postoperative IOP control [38]. However, in this study, no significant difference in IOP outcomes was observed between the two surgical techniques. Therefore, the choice of surgical method should be individualized based on patient-specific factors and the surgeon's expertise.

Based on real-world clinical data, this study systematically analyzed the clinical characteristics and short-term surgical outcomes of IOL dislocation, providing valuable data to support the optimization of clinical management pathways and the development of risk assessment models. Nevertheless, certain limitations remain. First, although pseudoexfoliation syndrome is considered a major cause of IOL dislocation in Western populations, it was not included in this analysis due to its low prevalence in the Central China region, which may limit the generalizability of the findings. Second, the retrospective observational design of this study carries inherent risks of selection bias and incomplete information. Future research should include large-scale, multicenter, prospective cohort studies that incorporate molecular biology and imaging techniques to further elucidate the mechanisms of IOL dislocation and explore more effective treatment strategies.

In conclusion, the findings of this study contribute to the early identification of high-risk patients, the refinement of follow-up protocols, and the development of individualized treatment plans. The results carry significant clinical and research value. Although IOL dislocation is relatively uncommon, its potential to cause severe visual impairment underscores the importance of long-term follow-up and comprehensive management for high-risk individuals.

# 5 Conclusion

Progressive zonular insufficiency is the leading cause of IOL dislocation, highlighting the need for early risk assessment and precise surgical management to optimize outcomes.

## Supporting information

**S1 File. Raw data and key statistical analysis procedures. This file contains the unprocessed raw data used in the study, the within-group rank sum test results, and the between-group Chi-square test procedures.**
(PDF)

## Author contributions

**Conceptualization:** Su Xu, Jingzhi Shao.

**Data curation:** Su Xu, Jingzhi Shao.

**Methodology:** Su Xu, Yuhang Zhang, Wei Si.

**Project administration:** Fengyan Zhang.

**Writing – original draft:** Su Xu.

**Writing – review & editing:** Yi Mao, Shanshan Du, Fengyan Zhang.

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
