## [Decision Letter · Decision Letter 0]

Clinical Characteristics and Outcomes of Intraocular Lens Dislocation: An Eight-Year Retrospective Study

PLOS ONE

Dear Dr. Zhang,

Thank you for submitting your manuscript to PLOS ONE. After careful consideration, we feel that it has merit but does not fully meet PLOS ONE’s publication criteria as it currently stands. Therefore, we invite you to submit a revised version of the manuscript that addresses the points raised during the review process.

We look forward to receiving your revised manuscript.

Kind regards,

Koichi Nishitsuka

Academic Editor

PLOS ONE

Journal Requirements:

3. We note that your Data Availability Statement is currently as follows: All relevant data are within the manuscript and in Supporting Information files.

5. Please ensure that you refer to Figure 1-2 in your text as, if accepted, production will need this reference to link the reader to the figure.

6. We note you have included a table to which you do not refer in the text of your manuscript. Please ensure that you refer to Table 1 in your text; if accepted, production will need this reference to link the reader to the Table.

Additional Editor Comments:

1. Abbreviation "UCVA"

The manuscript uses the term “UCVA” for uncorrected visual acuity. While this abbreviation is still used in some contexts, the internationally preferred and more precise term is “UDVA” (Uncorrected Distance Visual Acuity), especially when referring to postoperative outcomes related to distance vision.

If the authors intentionally chose to use “UCVA,” we kindly ask that you clarify this in the Methods section or Abbreviations list, and ensure consistent usage throughout the manuscript. Otherwise, we recommend revising to “UDVA” for clarity and consistency with current standards.

2. Inclusion and Exclusion Criteria

The inclusion criteria for IOL dislocation are vague. Please clarify how dislocation was defined—e.g., complete dislocation versus subluxation—and whether objective thresholds (such as decentration >1 mm or tilt >10°) were applied.

The exclusion criterion, “poor ocular condition rendering examination infeasible,” is also too general. Please specify which conditions were excluded (e.g., corneal opacity, phthisis bulbi, trauma, etc.). A standardized and reproducible definition of inclusion/exclusion is essential.

3. Minimum Follow-up Period

No minimum postoperative follow-up duration is described. This is particularly important for cases included up to October 2024, which may not have had sufficient follow-up at the time of submission. Please state the minimum follow-up duration and confirm that all included cases met this criterion.

4. Timing of Visual Acuity Evaluation

The manuscript does not specify when postoperative visual acuity was measured (e.g., 1 week, 1 month, etc.). As visual recovery evolves over time, please provide the exact time points at which visual outcomes were assessed.

5. Lack of Stratified Outcome Analysis by Surgical Technique

The study includes multiple surgical approaches (e.g., IOL repositioning, exchange, removal), but outcomes are presented in aggregate. To enhance clinical utility, we recommend presenting visual acuity and complication rates stratified by surgical method.

6. Method for Assessing Postoperative IOL Position

The manuscript mentions that IOLs were in a “normal position” postoperatively, but the method for determining this is unclear. Was it based on slit-lamp examination alone, or were imaging modalities (e.g., anterior segment OCT or UBM) used? Please clarify.

7. Statistical Methodology

While the authors state that normality was tested, the method (e.g., Shapiro–Wilk) and results are not provided. Please specify how normality was assessed and confirm that appropriate statistical tests were used based on the distribution of data.

Additionally, given the dataset size (166 eyes), we encourage the use of multivariate analysis (e.g., logistic regression) to adjust for potential confounding factors (e.g., age, high myopia, prior vitrectomy, surgical method). This would greatly strengthen the validity of the conclusions.

Reviewers' comments:

Reviewer's Responses to Questions

**Comments to the Author**

1. Is the manuscript technically sound, and do the data support the conclusions?

Reviewer #1: Yes

2. Has the statistical analysis been performed appropriately and rigorously?

Reviewer #1: Yes

3. Have the authors made all data underlying the findings in their manuscript fully available?

Reviewer #1: Yes

4. Is the manuscript presented in an intelligible fashion and written in standard English?

Reviewer #1: Yes

Reviewer #1: The authors present an interesting retrospective study of 166 eyes with IOL dislocation.

English grammar and writing can be improved.

I would like to clarify some issues:

1. Abstract. The authors state that ‘no severe complications were observed, except for elevated intraocular pressure, which occurred in 11.04 percent of cases.’ Was IOP increase severe in 11.04%?

2. Abstract. According to the authors, ‘Postoperative visual acuity improved significantly postoperative visual acuity improved significantly’. Do you mean UDVA?

3. Abstract. The authors state that ‘no severe complications were observed.’ However, 12 eyes were not operated on because of ‘concerns about surgical outcomes’. In addition, 12 eyes did not receive an IOL. In my opinion, the sentence ‘no severe complications were observed’ is misleading.

4. Methods. Subjects. Three items appear before the exclusion criteria. Were they the inclusion criteria of the study?

5. Methods. Ophthalmological examinations. The authors state that they used IOL Master 500/700 but they do not explain how the IOL power was chosen.

6. Methods. IOL removal. Please explain why some eyes were not suitable to receive, for example, an iris-fixated IOL (eg. Artisan).

7. Results. Comorbid conditions. Please explain what ‘glaucoma’ is. There are different types of glaucoma (eg. pseudoexfoliative glaucoma).

8. Results. Comorbid conditions. Cataract appears with age, as hypertension, stents, cholecystectomy and diabetes. Therefore, these conditions are not rare in cataract patients. Please compare the incidence of both diseases in the group of patients with IOL dislocation and the control patients. Same with the ‘unhealthy habits.’

9. Results. Comorbid conditions. Macular edema may appear due to diabetes or inflammation. Please explain this issue.

10. Results. Please use UDVA (uncorrected distance visual acuity) instead of the old-fashioned UCVA.

11. Results. Although UDVA is interesting, corrected distance visual acuity (CDVA) must be given if you want to present your clinical results. Moreover, in some cases, the IOL was changed and an IOL with better power was implanted. This is a very important limitation of this study.

**Do you want your identity to be public for this peer review?** For information about this choice, including consent withdrawal, please see our Privacy Policy

Reviewer #1: No

---

## [Author Response · Author response to Decision Letter 1]

25 May 2025

Response Letter to Reviewer Comments

We would like to thank the efforts of the editorial personnel and the reviewers. Comments are in black, our responses are in blue, and quotes from the manuscript or details are in purple.

SECTION 1 Journal Requirements

Response: We have checked the style requirements of PLOS ONE, including document naming requirements, to ensure that the manuscript meets the requirements.

Response: We have provided additional details regarding the participant's consent, ensured that the type of informed consent and full anonymization of the data were specified, and provided the ethics approval number in the Methods section (Line 87-90).

All patient records and data were anonymized and de-identified. The study was approved by the institutional ethics committee (2022-KY-0006-001), and written informed consent was obtained from all participants. In the case of minors, written consent was obtained from their guardians.

3. We note that your Data Availability Statement is currently as follows: All relevant data are within the manuscript and in Supporting Information files.

Response: Specific deidentified data sets have been made available in the Supplementary file, but there may be ethical or legal restrictions on public access because they contain potentially sensitive patient information.

Response: We have removed the ethics statement except for the Methods section.

5. Please ensure that you refer to Figure 1-2 in your text as, if accepted, production will need this reference to link the reader to the figure.

Response: We have referenced Figures 1-2 correctly in the text.

6. We note you have included a table to which you do not refer in the text of your manuscript. Please ensure that you refer to Table 1 in your text; if accepted, production will need this reference to link the reader to the Table.

Response: We have referenced Table 1 correctly in the text.

Response: We have included the title of the supporting information at the end of the manuscript and updated the corresponding textual citations.

SECTION 2 Additional Editor Comments

1. Abbreviation "UCVA"

The manuscript uses the term “UCVA” for uncorrected visual acuity. While this abbreviation is still used in some contexts, the internationally preferred and more precise term is “UDVA” (Uncorrected Distance Visual Acuity), especially when referring to postoperative outcomes related to distance vision.

If the authors intentionally chose to use “UCVA,” we kindly ask that you clarify this in the Methods section or Abbreviations list, and ensure consistent usage throughout the manuscript. Otherwise, we recommend revising to “UDVA” for clarity and consistency with current standards.

Response: Thank you for your valuable advice. We have revised the "UCVA" to "UDVA" in the full text.

2. Inclusion and Exclusion Criteria

The inclusion criteria for IOL dislocation are vague. Please clarify how dislocation was defined—e.g., complete dislocation versus subluxation—and whether objective thresholds (such as decentration >1 mm or tilt >10°) were applied.

The exclusion criterion, “poor ocular condition rendering examination infeasible,” is also too general. Please specify which conditions were excluded (e.g., corneal opacity, phthisis bulbi, trauma, etc.). A standardized and reproducible definition of inclusion/exclusion is essential.

Response: Thank you for your advice. We have clarified the objective threshold for dislocation adoption and added specifics to the exclusion criteria in the inclusion section of the Methods section (Line 69-86).

Inclusion criteria:

(1) History of IOL implantation;

(2) Dislocation of the IOL or the IOL–capsular bag complex, confirmed by slit-lamp biomicroscopy, ophthalmoscopy, ultrasound biomicroscopy (UBM), or optical coherence tomography (OCT), defined as dislocation >0.5 mm from the original position or tilt angle >7° following adequate mydriasis with tropicamide;

(3) Complete clinical and follow-up data, including detailed preoperative, intraoperative, and postoperative records, and at least one day of postoperative follow-up;

(4) Normal cognitive function and the ability to cooperate with ocular examinations.

Exclusion criteria:

(1) Severe corneal pathology, endophthalmitis, or other uncontrolled ocular conditions preventing necessary examinations;

(2) Incomplete clinical data, inadequate postoperative follow-up, or failure to comply with follow-up plans resulting in missing key pre- or postoperative data.

3. Minimum Follow-up Period

No minimum postoperative follow-up duration is described. This is particularly important for cases included up to October 2024, which may not have had sufficient follow-up at the time of submission. Please state the minimum follow-up duration and confirm that all included cases met this criterion.

Response: Thank you for your advice. We have described the minimum duration of follow-up in the Methods section and confirmed that all enrolled cases met this criterion(Line 76-78).

Complete clinical and follow-up data, including detailed preoperative, intraoperative, and postoperative records, and at least one day of postoperative follow-up.

4. Timing of Visual Acuity Evaluation

The manuscript does not specify when postoperative visual acuity was measured (e.g., 1 week, 1 month, etc.). As visual recovery evolves over time, please provide the exact time points at which visual outcomes were assessed.

Response: Thank you for your valuable advice. We have described the time of postoperative visual acuity measurement (1 day after surgery) in the Methods section.

Complete clinical and follow-up data, including detailed preoperative, intraoperative, and postoperative records, and at least one day of postoperative follow-up.

5. Lack of Stratified Outcome Analysis by Surgical Technique

The study includes multiple surgical approaches (e.g., IOL repositioning, exchange, removal), but outcomes are presented in aggregate. To enhance clinical utility, we recommend presenting visual acuity and complication rates stratified by surgical method.

Response: Thank you for your valuable advice. We have shown visual acuity and complication rates stratified by different surgical methods in the results section(Line233-249).

Table 4 Comparison of UDVA between different surgical techniques for IOL dislocation*.

Surgical technique Preoperative UDVA Postoperative UDVA Z P

IOL repositioning�n=100� 0.70�0.30�1.28� 1.50�1.00�2.00� -3.187 0.001

IOL replacement�n=39� 0.40�0.80�1.30� 1.50�1.00�2.00� -2.159 0.031

IOL explantation�n=12� 1.35�0.94�2.00� 1.50�1.00�2.00� -0.352 0.725

6. Method for Assessing Postoperative IOL Position

The manuscript mentions that IOLs were in a “normal position” postoperatively, but the method for determining this is unclear. Was it based on slit-lamp examination alone, or were imaging modalities (e.g., anterior segment OCT or UBM) used? Please clarify.

Response: Thank you for your advice. We have clarified in the text that the postoperative IOL position was based on slit-lamp examination (Line241-242).

On postoperative day one, slit-lamp examination confirmed that the IOLs were well-centered, without decentration or tilt.

7. Statistical Methodology

While the authors state that normality was tested, the method (e.g., Shapiro–Wilk) and results are not provided. Please specify how normality was assessed and confirm that appropriate statistical tests were used based on the distribution of data.

Additionally, given the dataset size (166 eyes), we encourage the use of multivariate analysis (e.g., logistic regression) to adjust for potential confounding factors (e.g., age, high myopia, prior vitrectomy, surgical method). This would greatly strengthen the validity of the conclusions.

Response: Thank you for your valuable advice. We have specified how normality is assessed and have used appropriate statistical tests according to the distribution of the data in the text(Line158-166).

Based on your suggestions, we will further use Logistic regression and other methods to adjust potential confounding factors in future case-control studies.

Statistical analysis was performed using SPSS version 27.0. Data distribution was assessed using the Kolmogorov–Smirnov or Shapiro–Wilk test. Normally distributed continuous variables were expressed as mean ± standard deviation (SD) and compared using independent samples t-tests. Non-normally distributed data were expressed as median and interquartile range (IQR), and compared using nonparametric rank-sum tests. Categorical variables were presented as frequencies and percentages and compared using the χ² test. A P-value <0.05 was considered statistically significant.

SECTION 3 Reviewers' comments

The authors present an interesting retrospective study of 166 eyes with IOL dislocation.

English grammar and writing can be improved.

Response: Thank you for your valuable advice. We have polished the full text again.

I would like to clarify some issues:

1. Abstract. The authors state that ‘no severe complications were observed, except for elevated intraocular pressure, which occurred in 11.04 percent of cases.’ Was IOP increase severe in 11.04%?

Response: Thank you for your advice. We have provided further clarification in the abstract section. The postoperative elevation of intraocular pressure in all patients was transient and controllable, and returned to normal within 3 days after intervention, so it was not severe(Line 29-31).

Postoperative transient intraocular pressure elevations were observed without significant differences between surgical methods, resolving within three days in all cases.

2. Abstract. According to the authors, ‘Postoperative visual acuity improved significantly postoperative visual acuity improved significantly’. Do you mean UDVA?

Response: Thank you for your advice. Postoperative visual acuity refers to the UDVA(Line 26-28). We have provided further clarification in the abstract section.

Of the 154 eyes that underwent surgical intervention, both IOL repositioning and exchange procedures resulted in significant improvements in UDVA (P < 0.05).

3. Abstract. The authors state that ‘no severe complications were observed.’ However, 12 eyes were not operated on because of ‘concerns about surgical outcomes’. In addition, 12 eyes did not receive an IOL. In my opinion, the sentence ‘no severe complications were observed’ is misleading.

Response: Thank you for your valuable advice. No severe complications were observed among patients who underwent surgical intervention. We originally expressed ambiguities, we have revised the abstract and clarified in the main text (Line 227-231).

Of the 166 eyes, 154 underwent surgical treatment. The remaining 12 eyes did not receive surgery due to low endothelial cell counts (<500 cells/mm²) or patient concerns about surgical outcomes (Table 3). All surgeries were completed successfully without intraoperative complications.

4. Methods. Subjects. Three items appear before the exclusion criteria. Were they the inclusion criteria of the study?

Response: Thank you for reminding us that we lost the "inclusion criteria".What appeared before the exclusion criteria were the inclusion criteria for the study, which we have clarified in the main text(Line 69-80).

Inclusion criteria:

(1) History of IOL implantation;

(2) Dislocation of the IOL or the IOL–capsular bag complex, confirmed by slit-lamp biomicroscopy, ophthalmoscopy, ultrasound biomicroscopy (UBM), or optical coherence tomography (OCT), defined as dislocation >0.5 mm from the original position or tilt angle >7° following adequate mydriasis with tropicamide;

(3) Complete clinical and follow-up data, including detailed preoperative, intraoperative, and postoperative records, and at least one day of postoperative follow-up;

(4) Normal cognitive function and the ability to cooperate with ocular examinations.

5. Methods. Ophthalmological examinations. The authors state that they used IOL Master 500/700 but they do not explain how the IOL power was chosen.

Response: Thank you for your advice. Because our study included 8 years of clinical data, during which the IOL Master 700 was introduced, before the introduction of the IOL Master 700 device, all data were collected with the IOL Master 500.

6. Methods. IOL removal. Please explain why some eyes were not suitable to receive, for example, an iris-fixated IOL (eg. Artisan).

Response: Thank you for your advice. Combined with our preoperative examination, we thought that the patient had the risk of corneal decompensation or visual acuity still not improved after IOL implantation. After communicating with the patient and his family, the patient and his family voluntarily chose not to accept it. We consider the implantation of an IOL to be an additional financial burden to the patient and his family in a situation where vision cannot be salvaged.

7.

---

## [Editor Report · Decision Letter 1]

Clinical characteristics and outcomes of intraocular lens dislocation: an eight-year retrospective study

PONE-D-25-02020R1

Dear Dr. Zhang,

We’re pleased to inform you that your manuscript has been judged scientifically suitable for publication and will be formally accepted for publication once it meets all outstanding technical requirements.

Kind regards,

Koichi Nishitsuka

Academic Editor

PLOS ONE
---

## [Editor Report · Acceptance letter]

PONE-D-25-02020R1

PLOS ONE

Dear Dr. Zhang,

I'm pleased to inform you that your manuscript has been deemed suitable for publication in PLOS ONE. Congratulations! Your manuscript is now being handed over to our production team.

Kind regards,

on behalf of

Dr. Koichi Nishitsuka

Academic Editor

PLOS ONE